# A Cross-Sectional Validation Study of Camry EH101 versus JAMAR Plus Handheld Dynamometers in Colorectal Cancer Patients and Their Correlations with Bioelectrical Impedance and Nutritional Status

**DOI:** 10.3390/nu16121824

**Published:** 2024-06-11

**Authors:** Andrés Jiménez-Sánchez, José Luis Pereira-Cunill, María Luisa Limón-Mirón, Amelia López-Ladrón, Francisco Javier Salvador-Bofill, Pedro Pablo García-Luna

**Affiliations:** 1Unidad de Gestión Clínica de Endocrinología y Nutrición, Instituto de Biomedicina de Sevilla, IBiS/Hospital Universitario Virgen del Rocío/CSIC/Universidad de Sevilla, Avda. Manuel Siurot s/n, 41013 Seville, Spain; garcialunapp@yahoo.es; 2Unidad de Gestión Clínica de Oncología Médica, Hospital Universitario Virgen del Rocío, Avda. Manuel Siurot s/n, 41013 Seville, Spain; marial.limon.sspa@juntadeandalucia.es (M.L.L.-M.); amelia.lopezladron.sspa@juntadeandalucia.es (A.L.-L.); franciscoj.salvador.sspa@juntadeandalucia.es (F.J.S.-B.)

**Keywords:** handgrip strength, JAMAR, Camry, Smedley, dynamometer, colorectal cancer, bioelectrical impedance, muscle mass, GLIM criteria, morphofunctional assessment

## Abstract

Background: Reduced muscle strength (dynapenia) and mass (atrophy) are prognostic factors in oncology. Measuring maximal handgrip strength with dynamometers is feasible but limited by the cost of the reference device (JAMAR). Methods: A cross-sectional study was conducted on colorectal cancer outpatients treated with chemotherapy or under active surveillance in our center from September 2022 to July 2023. Accuracy, reliability, and concordance were compared for two handheld dynamometers: the JAMAR Plus (the gold-standard device) and the Camry EH101 (a low-cost index device). A simultaneous nutritional diagnosis with GLIM criteria and bioelectrical impedance analysis (BIA) was carried out. Results: A total of 134 participants were included. The median of maximal strength for the JAMAR Plus had a non-significant difference of 1.4 kg from the Camry EH101. The accuracy and reliability of the devices were high. Bland–Altman analysis showed a 0.8 kg bias and −4.1 to 5.6 kg limits of agreement (LoA); a 0.1 kg bias and −5.3 to 5.4 kg LoA in men; a 1.5 kg bias and −2.2 to 5.3 kg LoA in women. In total, 29.85% of the participants were malnourished. Prevalence of dynapenia increased from 3.67% with the JAMAR Plus to 5.14% with the Camry EH101. Both devices had a moderate and significant correlation with BIA-estimated muscle mass. Conclusions: The Camry EH101 was a cost-effective alternative to JAMAR Plus in our sample.

## 1. Introduction

Nutritional screening, diagnosis, and intervention are of capital importance in oncology. One of the cornerstones of malnutrition is muscle mass (MM) loss (also known as muscle atrophy). Subject-related, tumor-related, and treatment-related characteristics can cause weight-independent low MM in oncology [1]. A previous study on colorectal cancer showed a prevalence of low MM in 39.8% of non-metastatic and surgically treated patients, with a prevalence of up to 46.2% in cases of metastatic disease [2]. MM loss acts as an independent predictor of survival [3,4]; physical, cognitive, and social functioning [5,6]; health-related quality of life [7]; postoperative complications and hospital length of stay [8,9]; and dose-limiting toxicity from capecitabine and bevacizumab [10] in colorectal cancer.

MM can be measured in clinical settings with indirect body composition methods, such as computed tomography (CT), magnetic resonance imaging, and ultrasound. MM can also be estimated using doubly indirect body composition methods, such as bioelectrical impedance analysis (BIA) and anthropometry.

BIA has widespread use in clinical nutrition [11] and few technical limitations in oncological outpatients [12]. The device emits a small current that passes through the human body, measuring the following raw bioelectrical parameters: resistance (R), capacitive reactance (X_c_), and phase angle (θ). R and X_c_ can be adjusted by height and depicted in relation to the confidence ellipses of a healthy reference population. This methodology is called bioimpedance vector analysis (BIVA) or “RX_c_ graph” [13], and it facilitates the visualization and monitoring of cellularity and tissue hydration independently of weight. An RX_c_ graph classifies subjects in four quadrants, top-down and clockwise: lean, cachectic, obese, and athletic. Regarding θ, it has a moderate correlation with CT-measured muscle atrophy and myosteatosis (fat infiltration in skeletal muscle) and has enough specificity to act as a surrogate for these conditions in patients with colorectal cancer [14]. It is also an independent risk factor for mortality in different cancers [15]. Additionally, θ can be transformed into a standardized phase angle (Z-score), which is also an independent predictor of mortality in oncology [16].

BIA also allows the estimation of MM and fat mass (FM) with regression equations, ideally specific to the clinical population under study [17]. BIA-estimated MM has shown a moderate correlation with CT results in cross-sectional studies in both non-malnourished and malnourished cancer patients [18,19]. In longitudinal studies, BIA-estimated MM has shown a predictive ability for survival similar to CT-measured MM in colorectal cancer [20].

As body composition analysis is not always available in clinical scenarios, low strength (dynapenia) can be used as a proxy for low MM due to its correlation. Therefore, it is included in operative definitions of malnutrition, such as the GLIM criteria [21]. In addition, dynapenia is an independent predictor of lower quality of life [22], more postoperative complications [23], and worse survival [24,25] in colorectal cancer.

In contrast to laboratory methods, measurement of maximal handgrip strength via handheld dynamometry is a feasible way to determine dynapenia in clinical practice in outpatients with malignancies [26] and other clinical populations in routine care. Traditionally, the gold-standard device for this task in clinical nutrition is the JAMAR hydraulic handheld dynamometer with analogue readout. Nevertheless, there is a new version (the JAMAR Plus) with digital readouts that has substituted the hydraulic cells with electronic load sensors. Smedley-type spring handheld dynamometers with digital readouts, such as Baseline, Medline, and Camry devices, are alternatives to JAMAR instruments which also have survival prediction ability [27] and normative values [28]. Their main pros are their lower price, lower weight (356 g for the Camry device versus 727 g for the JAMAR Plus), and easier readability (when compared to an analogue JAMAR dynamometer).

However, there appear to be important differences in the measurement capabilities of different types of handheld dynamometers [29] and between Smedley-type and JAMAR-type handheld dynamometers [30]. To our knowledge, five studies [31,32,33,34,35] have previously carried out a cross-sectional validation of the use of the Camry dynamometer as an index technique against the use of the analogue JAMAR dynamometer as a reference technique in different clinical populations. Our study aims to cross-sectionally validate the use of a low-cost digital spring handheld dynamometer (a Camry device) as an index technique against the use of a digital electronic load sensor handheld dynamometer (the JAMAR Plus) as a reference technique for the measurement of handgrip strength. This study relied on a single operator that measures outpatients with digestive neoplasms in our center. Secondarily, we describe the clinical situation and BIA-derived body composition of patients, analyzing the degree of correlation of both handheld dynamometers with MM and a standardized phase angle.

## 2. Materials and Methods

### 2.1. Study Design

Study design: an analytical observational study. Measurement period: September 2022 to July 2023.

Subjects under study: inclusion criteria: people ≥ 18 years old, either under active chemotherapy treatment (defined as any treatment session in the last 30 days) or active surveillance by oncology due to a documented diagnosis of colorectal cancer, with an ECOG of 0–3 and who completed a specific informed consent protocol. Exclusion criteria: medical ward inpatients, ECOG 4–5, surgery or intercurrent illness leading to hospital admission in the previous 30 days, pacemaker or implantable cardioverter defibrillator (ICD), skin lesions or severe adhesive dermatitis contraindicating the use of electrodes, consumption of diuretics in the last 48 h, severe cognitive impairment, or end-of-life care. Withdrawal criterion: at the request of the participant.

Sampling: convenience sampling with consecutive inclusion. Sample size: following the methodology proposed by Lu et al. [36] and using the statistical package *blandPower* (https://rdrr.io/github/nwisn/blandPower/f/README.md, accessed on 2 June 2024) in Rstudio software (version 2023.06.1+524), we determined a priori a risk of type I error = 0.05 and a risk of type II error = 0.20. Taking the minimum detectable change in mean handgrip strength of 5.44 kg from Huang et al. [33], we increased it by 40% and obtained 7.6 kg, which we established as the maximal clinically acceptable difference between devices (delta). On the other hand, we took a mean absolute difference of 0.1 kg and calculated a standard deviation of the difference of 3.1 kg, according to Andrade et al. [35]. With these data, we obtained a sample size of *n* = 132 pairs of measurements (and thus participants).

The study was conducted in accordance with the Declaration of Helsinki and approved by the Ethics Committee “CEI de los Hospitales Universitarios Virgen Macarena y Virgen del Rocío” (protocol code: 1006-N-22; date of approval: 23 May 2022).

### 2.2. Data Collection

#### 2.2.1. Clinical Variables and Cancer Staging

The following clinical variables were obtained from digitized health records (“DIRAYA Clinical Station”): age, ECOG performance status [37], type of baseline disease, tumor staging (according to AJCC-TNM, 8th edition) [38], type and date of treatment. Different systemic anticancer therapies were applied following standard schemes according to oncology guidelines [39].

#### 2.2.2. Handgrip Strength Protocol

Handgrip strength was measured in kilograms (kg) using randomly and consecutively a JAMAR Plus handheld dynamometer (Performance Health Sammons Preston, Warrenville, IL, USA) and a Camry EH101 handheld dynamometer (Zhongshan Camry Electronic Co. Ltd., Zhongshan, China). Both devices were purchased new and used exclusively for this study. All men were measured using grip position 2 of the JAMAR Plus [40,41] and an equivalent grip size of the Camry device. Women with smaller hands had their handgrip size adapted as previously described [41] for both handheld dynamometers. Hand width (in cm) was measured as the distance between the distal end of the thumb and the little finger at maximal abduction on a horizontal surface using a flexible and inextensible tape measure produced by CESCORF (Porto Alegre, Brazil). Strength was measured exclusively in the dominant hand (either right or left upon interrogation), following the Southampton protocol [42]. First, the initial device was randomly assigned. Then, a pre-test measurement was performed in all cases with each device to familiarize the participant with both devices. Then, three measurements were alternately (1:1) obtained with each device. Each registered attempt consisted of a maximal contraction for at least three seconds [43], allowing a 60 s rest between each attempt [44]. The participant was not informed which of the devices was considered the gold standard. Both the mean and the maximum values of handgrip strength were used for analysis.

#### 2.2.3. Operative Definitions of Dynapenia, Malnutrition, and Sarcopenia

To diagnose dynapenia, maximal handgrip strength was compared to the maximal handgrip strength normative values developed by Dodds et al. [45] from population-based studies with JAMAR and Smedley-type dynamometers. These values are used in the European Working Group on Sarcopenia in Older People (EWGSOP-II) criteria [46]. Alternatively, the NAKO study percentiles [47] for the JAMAR Plus handheld dynamometer were used. Dynapenia was defined both with a percentile-based approach (maximal strength below the corresponding 10th percentile based on age and sex) and a dichotomous approach using EWGSOP-II cutoff values (<27 kg in men; <16 kg in women) [46].

Malnutrition was defined using the GLIM criteria [48]. Regarding the cutoff point for atrophy, we used the SMI cutoff points for the Spanish population of Masanés et al. [49], following EWGSOP-II [46] and GLIM [50] recommendations. We considered only unintentional weight loss when calculating GLIM scores. Sarcopenia was defined using the EWGSOP-II criteria [46].

#### 2.2.4. Bioelectrical Impedance Analysis Protocol

Bioelectrical impedance was measured with a phase-sensitive touchscreen impedance device (Nutrilab™, Akern, Florence, Italy), working with an alternating sinusoidal electric current of 230 μA at an operating frequency of 50 kHz (±1%). The device was calibrated every morning using the standard control circuit supplied by the manufacturer with a known impedance resistance (R) = 380 Ω and reactance (X_c_) = 45 Ω. Impedance data were shown directly on an LCD touchscreen device and stored in internal memory. Accuracy: Rz: ±0.1 Ω; X_c_: ±0.1 Ω; CV% < 1%. The pure bioelectric parameters obtained were Z (impedance, in Ω), R (resistance, in Ω), X_c_ (reactance, in Ω), and phase angle (θ = tan^−1^ (X_c_/R), in °). The measurement technique accorded with the ESPEN protocol [51]. After ensuring fasting from solids for at least eight hours, fasting from liquids for at least two hours, emptying of the bladder within 15 min prior to the test, and removal of metallic (jewelry) and electromagnetic devices (smartphones and other wearables), the participants were placed in a supine position and static electricity was removed. The participants’ arms were separated at 30° and their legs at 45°. The electrode placement area was cleaned with 70° alcohol. Once dry, two sets of adhesive Ag/AgCl low-impedance electrodes (Bivatrodes™ Akern Srl; Florence, Italy) designed for accurate and sensitive bioimpedance measurements were placed proximally to the phalangeal–metacarpal joint on the dorsal surface of the right hand and distally to the transverse arch on the superior surface of the right foot. Sensor electrodes were placed at the midpoint between the distal prominence of the radius and ulna of the right wrist and between the medial and lateral malleoli of the right ankle. The clamps of the measuring cable were then attached, avoiding the occurrence of loops. After an approximate time of five minutes in supine position, the participants underwent three consecutive measurements, and each R, X_c_, and θ value was registered. Janssen et al.’s equation [52] was used to estimate skeletal muscle mass (SMM). The Skeletal Muscle Index (SMI) was calculated as SMI = SMM/H^2^, where H = height (m). θ values were transformed into Z-scores according to the reference values provided by Bosy-Westphal et al. [53]. BIVA was performed according to Piccoli et al. [13], as previously described. Standardized phase angle values below −1.65 SD were defined as pathological, as previous studies have shown their predictive value in oncological populations [54].

#### 2.2.5. Basic Anthropometry Protocol

Height and weight measurements were performed according to ESPEN recommendations [51]. Height was measured with an SECA wall-mounted measuring rod (seca GmbH & Co. KG, Hamburg, Germany) installed at 200 mm. Weight was measured using a new and calibrated portable weight measurement device, the CAS PB-150 (CAS Corporation, Seoul, Republic of Korea), with a capacity of up to 150 kg and two ranges of measuring sensitivity (20 g differences up to 60 kg; 50 g differences between 60 and 150 kg).

#### 2.2.6. Data Quality

Data quality: all field measurements were carried out by a single researcher with experience in body composition analysis (A.J.S.). Following QUADAS-2 recommendations [55], body composition analysis and determination of handgrip strength were performed within 15 min of each other and by a single examiner, with the participant blinded to which handheld dynamometer was the gold standard. Cancer stagings, treatments, and performance scores were registered in the database as recorded by oncologists (M.L.L.-M. and A.L.-L.) in health records.

### 2.3. Data Analysis

Statistical analysis: RStudio software (version 2023.06.1+524) and the packages *tidyverse*, *psych*, *cowplot*, *irr*, *ggpubr*, and *Rcmdr* were used. Descriptive statistics of the variables of interest were determined. The presence or absence of normality was analyzed with the Shapiro–Wilk test. Variables with a normal distribution were described as means and standard deviations (SDs). Non-normally distributed variables were described as medians and interquartile ranges (IQRs). Next, we compared measures of central tendency (a comparison of means with a *t*-test in case of a normal distribution according to the Shapiro–Wilk test and homoscedasticity, and a comparison of medians with the Wilcoxon signed-rank test otherwise) of maximal and average handgrip strength with both handheld dynamometers (JAMAR Plus and Camry). Proportions were tested with *X*^2^ tests, using Yates’s continuity correction if *n_i_* < 5. The presence of random or systematic bias and the magnitude of agreement was determined by a Bland–Altman analysis. Limits of agreement (LoA) were defined with 95% confidence as $\bar{d}\;\pm 1.96\;SD$, where *d* = the mean of the difference between the methods and SD = the standard deviation of the difference between the methods [56]. The relative reliability of each instrument was assessed with the intraclass correlation coefficient (ICC), considering values of 0.75–0.90 as a “good” correlation and 0.91–1.00 as an “excellent” correlation [57]. As this study had a single clinical examiner and we wanted to extrapolate our results to routine clinical practice, we chose a two-way randomized model or ICC (2,1), according to Shrout and Fleiss [58], with the following formula: $\frac{MSr-MSe}{MSr+\left(K-1\right)\;MSe+\frac{K}{n}\;(MSc-MSe)}$, where MSr = the mean square of rows, MSe = the mean square of error, MSc = the mean square of columns, *k* = the number of measurements with the instrument, and *n* = the number of subjects. We assessed the absolute reliability of each instrument by the standard error of measurement (SEM) and its minimum detectable change (MDC), as previously described [33]. Simple correlation was calculated via the Pearson correlation coefficient. Statistical significance was determined in all tests as a *p*-value < 0.05.

## 3. Results

### 3.1. General Characteristics of the Sample

A total of *n* = 134 participants were measured, and their clinical and demographic characteristics are shown in Table 1. All subjects were included for statistical analysis.

### 3.2. Handgrip Strength Determined with Handheld Dynamometers

The entire sample was analyzed in the cross-sectional validation of the dynamometers. No participants had undergone previous surgery or had intercurrent diseases that may have affected their hand biomechanics.

#### 3.2.1. Correlation and Precision

The correlation between the maximal handgrip strengths determined by the JAMAR Plus and Camry devices was very high (r = 0.969; *p* < 2.2 × 10^−16^). The correlation between the average handgrip strengths determined by the JAMAR Plus and Camry devices was also very high (r = 0.974; *p* < 2.2 × 10^−16^). A graphical representation of this correlation analysis is shown in Figure 1.

The main variables related to the handheld dynamometers are summarized in Table 2. There was a prevalence of right-handed participants, and the men had bigger dominant hands than the women. *n* = 63 women and all the men performed handgrip strength measurements with the JAMAR Plus grip in position 2 and a Camry equivalent. Both maximal and average strength with both handgrip dynamometers were higher in men than in women. Women—but not men—had a greater difference in maximal strength between handgrip dynamometers.

#### 3.2.2. Comparison of Maximal Handgrip Strength between Devices

Maximal strength for both the JAMAR Plus and Camry handheld dynamometers followed a non-normal distribution. The median maximal strength for the JAMAR Plus dynamometer was 33.4 kg and the median maximal strength for the Camry dynamometer was 32.0 kg, with a non-significant difference of +1.4 kg in favor of the JAMAR Plus. A graphical representation is shown in Figure 2A. To improve comparability with previous studies, maximal strength for the JAMAR Plus averaged 34.5 kg (SD = 9.4 kg) and maximal strength for the Camry device averaged 33.7 kg (SD = 10.1 kg). When grouped by sex, the median maximal strength was 40.0 kg for males and 28.4 kg for females with the JAMAR Plus, while the median maximal strength was 39.0 kg for males and 27.4 kg for females with the Camry device. There were non-significant differences in men (+1.0 kg in favor of the JAMAR Plus) and women (+1.0 kg in favor of the JAMAR Plus). A graphical representation is shown in Figure 2B.

#### 3.2.3. Comparison of Average Handgrip Strength between Devices

The average strength for both the JAMAR Plus and Camry devices followed a non-normal distribution. The median average strength for the JAMAR Plus was 32.3 kg, and the median average strength for the Camry device was 31.0 kg. There was a non-significant difference of +1.3 kg in favor of the JAMAR Plus. A graphical representation is shown in Figure 3A. When divided by sex, the median of the average strength was 38.2 kg in males and 27.4 kg in females for the JAMAR Plus, while the median of average strength was 37.3 kg in males and 26.2 kg in females for the Camry device. There were non-significant differences in men (+0.9 kg in favor of the JAMAR Plus) and women (+1.2 kg in favor of the JAMAR Plus). A graphical representation is shown in Figure 3B.

#### 3.2.4. Distribution of Differences between Devices

Differences in maximal strength between the handheld dynamometers followed a normal distribution and are graphically represented in Figure 4A. When grouped by sex, differences in maximal strength were also normally distributed in both men and women and are graphically represented in Figure 4B.

#### 3.2.5. Bland–Altman Analysis

Bland–Altman analysis of the whole sample showed a 0.8 kg systematic bias in favor of the JAMAR Plus dynamometer, with limits of agreement of −4.1 to 5.6 kg. Only *n* = 6 measurements (4.47%) were outside the limits of agreement. A graphical representation is presented in Figure 5A. Bland–Altman analysis of the men showed a 0.1 kg systematic bias in favor of the JAMAR Plus dynamometer, with limits of agreement of −5.3 to 5.4 kg. Only *n* = 3 measurements (4.28%) were outside the limits of agreement. A graphical representation is presented in Figure 5B. Bland–Altman analysis of the women showed a 1.5 kg systematic bias in favor of the JAMAR Plus dynamometer, with limits of agreement of −2.2 to 5.3 kg. Only *n* = 2 measurements (3.12%) were outside the limits of agreement. A graphical representation is presented in Figure 5C.

#### 3.2.6. Differential Classification Bias in Dynapenia

Using the JAMAR Plus and the p10 cutoff points of Dodds et al. [45], there were *n* = 5 cases of dynapenia (all men), which were identical to *n* = 5 cases of dynapenia using the dichotomous EWGSOP cutoff points (all men). The prevalence of dynapenia rose to *n* = 21 cases when using the NAKO study’s cutoff points (*n* = 17 males, *n* = 4 females). Using the Camry device and the p10 cutoff points of Dodds et al., there were *n* = 7 cases of dynapenia (all males), while using the dichotomous cutoff points of EWGSOP-II, there were *n* = 6 cases of dynapenia (*n* = 5 males, *n* = 1 female). All *n* = 5 cases of JAMAR Plus dynapenia were included within the Camry dynapenia cases. The different prevalences of dynapenia provided by both handheld dynamometers are shown in Table 3. The increase in prevalence in absolute terms of dynapenia with the Camry dynamometer was +1.47% when using the p10 cutoff points of Dodds et al. and +0.74% when using the EWGSOP-II dichotomous cutoff points.

### 3.3. Nutritional Status Assessment

#### 3.3.1. Nutritional Diagnosis and Prevalence of Malnutrition

The prevalence of malnutrition in the sample was 29.85% (40/134), mainly at the expense of a 75.00% (30/40) diagnosis of muscle atrophy by BIA. The modal BMI category was overweight, followed by obesity. A detailed description of the parameters related to the GLIM nutritional diagnosis is given in Table 4.

#### 3.3.2. Body Composition Analysis with BIA

The most relevant BIA parameters for nutritional diagnosis grouped by sex are listed below in Table 5. Women had higher R and Z scores, and lower X_c_, θ, MM, and SMI scores. It is noteworthy that the disparity in the prevalence of muscle atrophy in women was much higher than in men (43.6% vs. 10.0%).

#### 3.3.3. Correlation between Maximal Strength and BIA-Estimated Muscle Mass

Maximal handgrip strength for the JAMAR Plus showed a moderate correlation with muscle mass (r = 0.639; *p* < 2.2 × 10^−16^) and SMI (r = 0.523; *p* = 1.075 × 10^−10^). Maximal handgrip strength for the Camry device also showed a moderate correlation with muscle mass (r = 0.667; *p* < 2.2 × 10^−16^) and SMI (r = 0.567; *p* = 1.141 × 10^−12^).

#### 3.3.4. Relations between Dynapenia, Muscle Atrophy, and Phase Angle

Using maximal strength as determined by the JAMAR Plus, there were *n* = 3 cases of dynapenia in the obesity BIVA quadrant and *n* = 2 cases in the cachexia BIVA quadrant. All cases of dynapenia determined by the JAMAR Plus were outside the confidence ellipse corresponding to p50 (see Figure 6A for a graphical representation). Using maximal strength as determined by the Camry device, there were *n* = 4 cases of dynapenia in the obesity BIVA quadrant, *n* = 2 cases in the cachexia BIVA quadrant, and *n* = 1 case of dynapenia in the lean BIVA quadrant. All cases of dynapenia determined by the JAMAR Plus device were outside the confidence ellipse corresponding to p50 (see Figure 6B for a graphical representation). A total of *n* = 35 participants had muscle atrophy (see Figure 6C for a graphical representation). No person with dynapenia had atrophy, and therefore there were no diagnoses of EWGSOP-II sarcopenia in the sample. A total of *n* = 32 participants (23.88%) displayed an unfavorable standardized phase angle. Using maximal handgrip strength as determined by the JAMAR Plus, all participants with dynapenia had an unfavorable standardized phase angle. Using maximal handgrip strength as determined by the Camry device, *n* = 5 participants with dynapenia had an unfavorable standardized phase angle and *n* = 2 participants with dynapenia had a normal standardized phase angle. We found that *n* = 7 participants with an unfavorable standardized phase angle also had muscle atrophy, while *n* = 28 participants with muscle atrophy had a normal standardized phase angle. A graphical representation is displayed in Figure 6D.

## 4. Discussion

The median maximal and mean strength distributions according to dynamometer type did not show statistically significant differences, although, in absolute terms, the JAMAR Plus showed a slightly higher maximal and mean strength than the Camry device. In comparison with previous studies, our differences are greater than those reported by Huang et al. [33] in *n* = 1064 community-dwelling adults in China and Tibet (65.13% female, 66 ± 7.7 years) with an average handgrip strength of 24.6 ± 8.1 kg for a JAMAR analogue device and 25.0 ± 7.8 kg for a Camry device. In fact, Andrade et al. [35] found no differences between devices in *n* = 220 elderly patients admitted for elective surgery in Chile (52.0% female, 73.1 ± 6.3 years) with an average handgrip strength of 26.9 ± 9.7 kg for a JAMAR analogue device and 26.9 ± 9.6 kg for a Camry device. On the other hand, our differences are smaller than those reported by Diaz-Muñoz et al. [31] in *n* = 133 community-dwelling adults in Colombia (50.4% female, 47 ± 20.74 years) with an average handgrip strength of 32.1 ± 9.9 kg for a JAMAR analogue device and 29.9 ± 9.2 kg for a Camry device.

Both handheld dynamometers displayed excellent validity, the values obtained being slightly higher than previous results [33]. Although the amplitudes of our limits of agreement in the Bland–Altman analyses were similar to or slightly smaller than those reported in previous research, the women in our sample had a small systematic bias in favor of the JAMAR Plus over the Camry device. These results differ from those reported by Huang et al. [33], where females had a bias of similar magnitude to males. We present a lower level of handgrip strength in the previous study or the use of a JAMAR analogue device instead of the JAMAR Plus device used in this study as hypothetical explanations of this finding. Interestingly, Savas et al. [44] found that, in people > 60 years, the JAMAR Plus overestimated handgrip strength in women by +1.4 kg compared to a JAMAR analogue device, this difference being negligible in men (−0.1 kg). These findings strikingly resemble the results we obtained when we compared the JAMAR Plus and Camry EH101 devices.

Although these handgrip strength differences were associated with an increased prevalence of dynapenia when using the index device (Camry) versus the reference device (JAMAR Plus) in our sample, we found this to be of no clinical significance due to the small absolute values. Nevertheless, this may not be the case in clinical populations with a high prevalence of dynapenia. Analogous to what was previously reported by the authors of the NAKO study [47], the use of their cutoff points for maximal strength as determined by the JAMAR Plus in our sample increased the prevalence of dynapenia from 3.67% to 15.44%. Prospective studies comparing the prognostic impact of using the Dodds et al. [45] cutoff points versus those of the NAKO study in different clinical populations will be necessary.

However, the prevalence of malnutrition in the sample was remarkable, despite the fact that most participants were under active surveillance. GLIM-based malnutrition was diagnosed mainly at the expense of muscle atrophy by BIA, which occurred primarily in women. There was no clear association between dynapenia and atrophy, with no cases of sarcopenia in the sample. Regarding the analysis of muscle atrophy, the association between muscle mass as determined by BIA and maximal strength was similar between the handheld dynamometers.

To our knowledge, the results of this research are unprecedented for three reasons. Firstly, we have not found a cross-sectional validation of the Camry dynamometer in an oncological population with colorectal cancer, which is of special interest due to its high incidence and prevalence. Secondly, previous cross-sectional validation studies were carried out with an analogue JAMAR device, which can yield different measurements to the JAMAR Plus device, although the magnitude of these differences varies between studies [29,44,59]. Finally, none of the previous studies performed with JAMAR analogue devices [31,32,33,34,35] performed a concomitant body composition analysis or tested whether Camry and JAMAR devices had similar or different associations with muscle mass in the study samples.

Regarding the main limitations of this study, the sampling was non-probabilistic, and the sample size calculation used a clinically acceptable difference between the devices that could be criticized as having been excessively lax, although the sample size was similar to [31] or larger [34] than those of some previous studies. Moreover, the results for the dynamometers could not be blinded at the time of data collection, as they were obtained by a single observer, although statistical analysis was performed after completion of the data collection. Although both devices were purchased specifically for this study and were factory-calibrated, the Camry dynamometer does not have—to our knowledge—a technical service for calibration and we do not know whether the long-term reliability of its measurements could vary with respect to those of the JAMAR Plus dynamometer, which does have an after-sales service for recalibration.

On the other hand, our design had several strengths. By using a single observer for the determination of strength and body composition, we eliminated a possible interobserver bias in both techniques. We used the gold-standard dynamometer that has classically been the gold standard in the field of clinical nutrition, and we eliminated a potential misclassification error by switching from an operator-dependent analogue reading to a digital one that is not subject to subjectivity and has a higher resolution (0.1 kg vs. 2.0 kg). We randomized the initial intervention and blinded the participants to which was the index technique and which was the reference technique, performing both at the same time. We used the same grip size for each individual regardless of the device and performed the Southampton protocol for measuring manual grip strength regardless of the type of dynamometer used, which we believe increased the internal validity of the study. In addition, all participants were included in the statistical analysis, and the same reference standard for the measurement of handgrip strength was used for all participants.

## 5. Conclusions

In our study, a Camry EH101 handheld dynamometer showed similar accuracy and reliability in comparison with a JAMAR Plus handheld dynamometer, although women showed a slight systematic bias and higher risk of false positives for dynapenia with the Camry device. Due to the lower cost and therefore the greater accessibility of the Camry device, we believe that the benefit of facilitating the diagnosis of dynapenia in resource-limited settings may outweigh the aforementioned risk of overdiagnosis of low maximal strength in a similar clinical population. The introduction of the Camry EH101 device in clinical units that usually have no access to handheld dynamometers, such as oncology, radiotherapy, surgery, and family medicine units, may facilitate early nutritional screening in colorectal cancer patients.

Future research ought to focus on transversal validation of the Camry device in other clinical populations at high nutritional risk (such as patients with other neoplasms or neurodegenerative or inflammatory diseases), using both a hydraulic JAMAR dynamometer and an electronic JAMAR Plus dynamometer as reference devices, stratifying data by sex and analyzing the impacts of conditions on dynapenia prevalence, as well as their correlations with muscle mass and other measurements of physical functioning. Additionally, longitudinal studies should be conducted to analyze the prognostic capacity of both devices for events of interest, such as survival rate, tolerance to treatments, and time to discharge, amongst others. Finally, other projects could analyze the durability, reliability, and accuracy of these devices in the long term.

## Figures and Tables

**Figure 1 nutrients-16-01824-f001:**
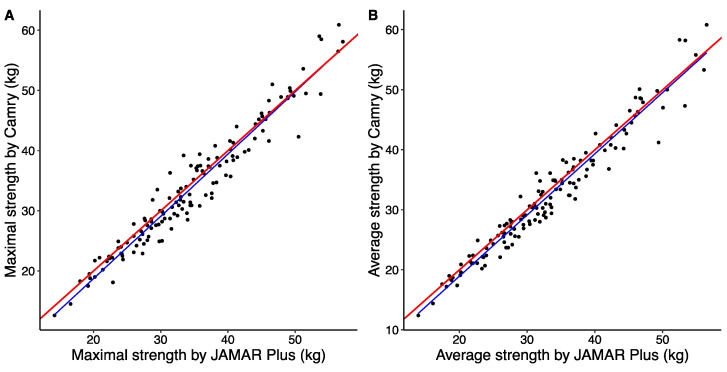
Simple linear regression of maximal handgrip strength (**A**) and average handgrip strength (**B**) between JAMAR Plus and Camry handheld dynamometers. The perfect bisector of interdevice regression is shown as a solid red line, and the linear regression model between the devices is shown as a solid blue line.

**Figure 2 nutrients-16-01824-f002:**
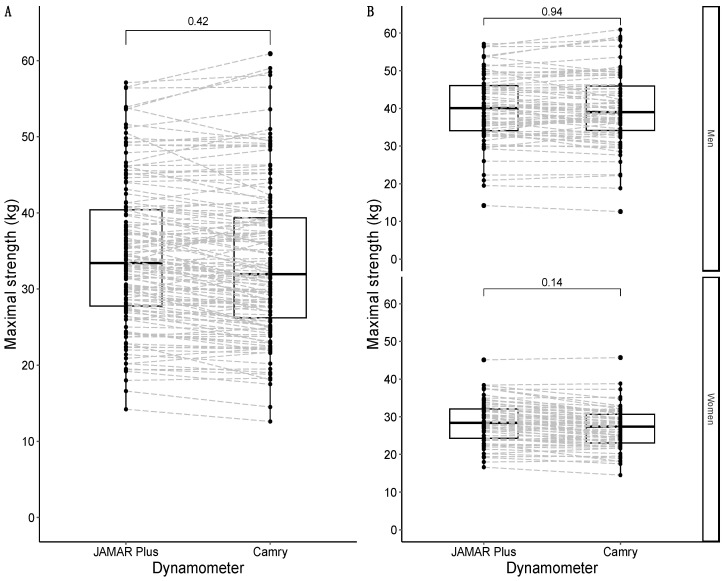
Box-and-whisker plots with overlapping geometries of points to represent the distribution of maximal strength in the sample: overall sample in (**A**); data grouped by sex in (**B**). Type of handheld dynamometer on the *X*-axis; maximal strength (in kg) on the *Y*-axis. The *p*-value of the median difference for maximal strength according to the type of dynamometer used (JAMAR Plus or Camry) is shown in square brackets. The dashed grey lines join the results of the same participants according to their identification numbers.

**Figure 3 nutrients-16-01824-f003:**
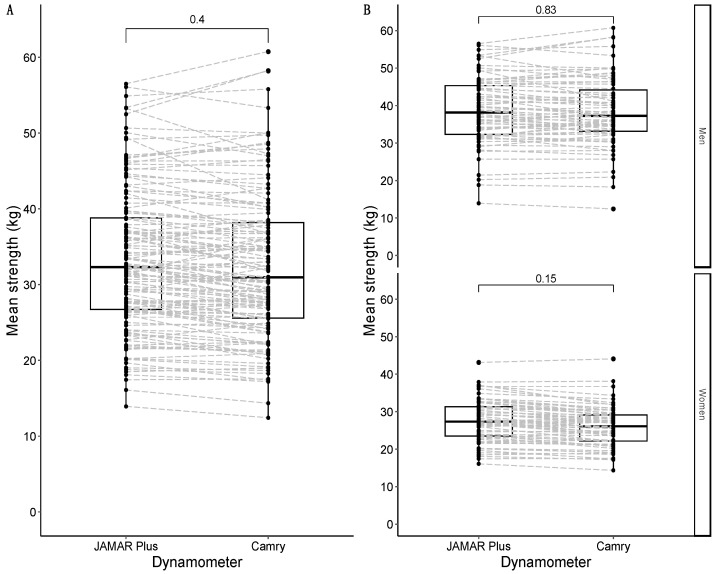
Box-and-whisker plots with overlapping geometries of points to represent the average strength distribution in the sample: overall sample in (**A**); data faceted by sex in (**B**). Type of handheld dynamometer on the *X*-axis; maximal strength (in kg) on the *Y*-axis. The *p*-value of the median difference for maximal strength according to the type of dynamometer used is shown in square brackets. The dashed grey lines join the results of the same participants according to their identification numbers.

**Figure 4 nutrients-16-01824-f004:**
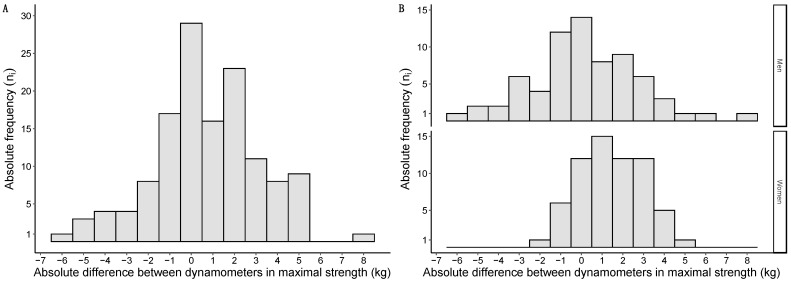
Histograms of the mean strength differences between dynamometers: overall sample in (**A**); data grouped by sex in (**B**). Absolute differences expressed in kg on the *X*-axis; absolute frequencies presented on the *Y*-axis.

**Figure 5 nutrients-16-01824-f005:**
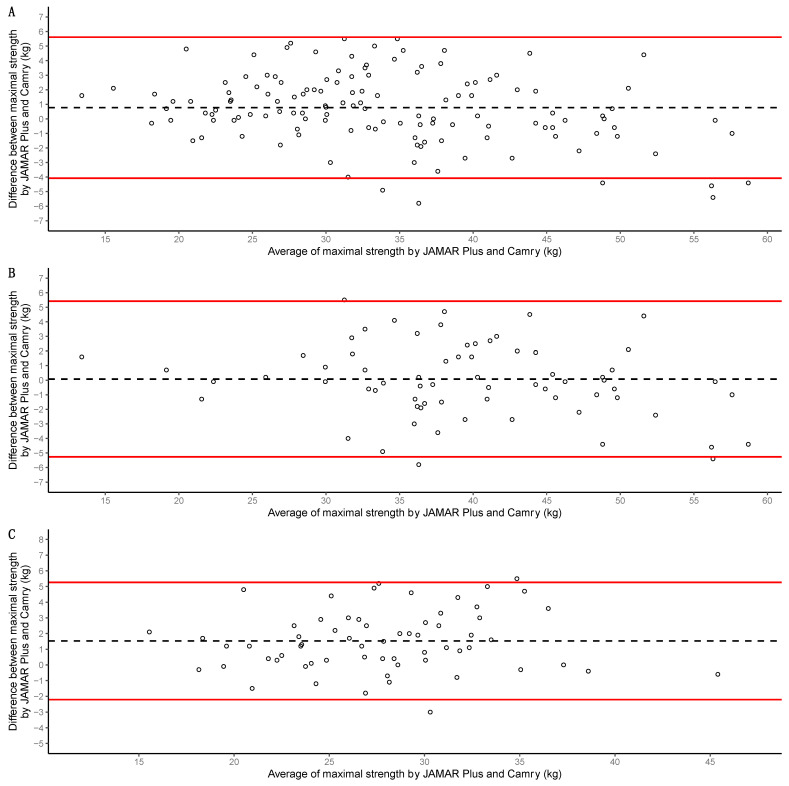
Bland–Altman analyses of the whole sample in (**A**), of the men in (**B**), and of the women in (**C**). Maximal handgrip strength for both devices (JAMAR Plus and Camry EH101) was used for calculations. Averages of maximal handgrip strength for both devices presented on the *X*-axis; differences in maximal handgrip strength for both devices presented on the *Y*-axis. Individual measurements are shown as empty circles; biases are represented as black dashed lines; upper and lower limits of agreement are shown as red solid lines.

**Figure 6 nutrients-16-01824-f006:**
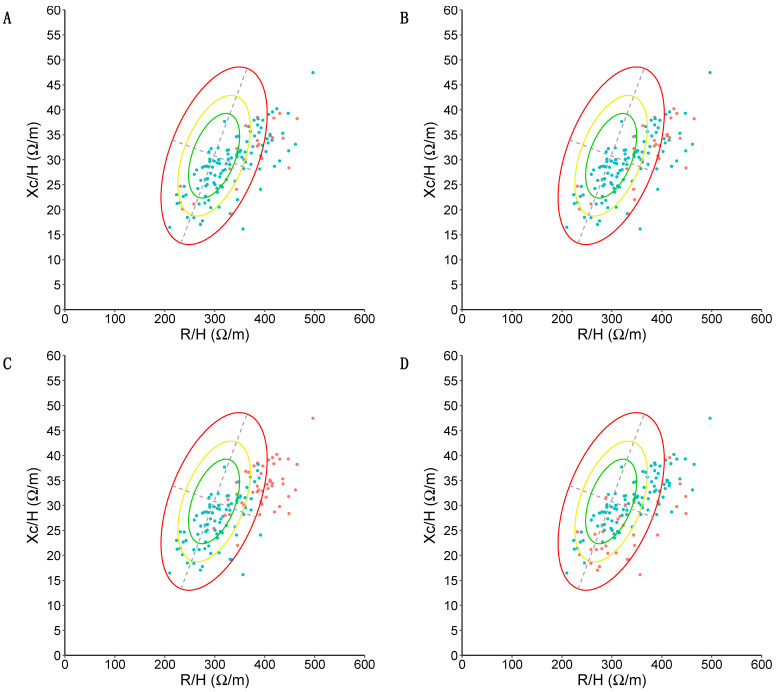
BIVA analysis of the sample. Resistance (R) divided by height (H) and expressed in ohm/meters on the *X*-axis; reactance (X_c_) divided by height (H) and expressed in ohm/meters on the *Y*-axis. Confidence ellipses corresponding to p50, p75, and p95 in the reference population are depicted in green (p50), yellow (p75), and red (p95). (**A**) Presence (red) or absence (green) of dynapenia is color-coded according to p10 values of Dodds et al. [45], according to maximal strength (in kg) as determined by the JAMAR Plus. (**B**) Presence (red) or absence (green) of dynapenia is color-coded according to p10 values of Dodds et al. [45], according to maximal strength (in kg) as determined by the Camry device. (**C**) Presence (red) or absence (green) of muscle atrophy is color-coded according to the Janssen et al. [52] equation with cutoff points from Masanés et al. [49]. Note how *n* = 33 (94.28% of people with muscle atrophy) appear in the lean quadrant and *n* = 2 appear in the cachexia quadrant. (**D**) Presence (red) or absence (green) of an unfavorable standardized phase angle (defined as < −1.65 standard deviations with respect to the reference population) is color-coded. Note how the majority of participants with an unfavorable phase angle are located in the cachexia quadrant (*n* = 18, 56.25%).

**Table 1 nutrients-16-01824-t001:** Clinical and demographic characteristics of the study sample.

Parameter	Results
Sample size (*n_i_*)	*n* = 134
Age (years)	Me = 65.7 years IQR = 13.4 years
Older than 65 (*n_i_*)	*n* = 71 (52.98%)
Female (*n_i_*)	*n* = 64 (47.76%)
Type of neoplasm (*n_i_*)	Right colon, *n* = 32 Rectum, *n* = 36 Sigma, *n* = 27 Left colon, *n* = 15 Transverse colon, *n* = 11 Rectosigmoid, *n* = 10 Krukenberg’s tumor (colon as the primary site), *n* = 1
Stage (TNM) at diagnosis	IIA (*n* = 24); IIB (*n* = 3); IIC (*n* = 4) IIIA (*n* = 12); IIIB (*n* = 43); IIIC (*n* = 13) IVA (*n* = 14); IVB (*n* = 15); IVC (*n* = 6)
Previous surgery	Yes, *n* = 123 No, *n* = 11
Active chemotherapy (*n_i_*)	No, *n* = 102 Yes, *n* = 32
ECOG (*n_i_*)	0, *n* = 86 1, *n* = 42 2, *n* = 5 3, *n* = 1

Clinical characteristics of the sample. Numerical values are expressed as absolute frequencies (*n*), percentages (%), medians (Me), and interquartile ranges (IQRs).

**Table 2 nutrients-16-01824-t002:** Strength measurements, precision, and reliability of the devices in the study sample.

Parameter	Total Sample	Men	Women	*p*-Value
Measuring side (*n_i_*)	Right, *n* = 125 Left, *n* = 9	Right, *n* = 65 Left, *n* = 5	Right, *n* = 60 Left, *n* = 4	<2.2 × 10^−16^
Hand size (cm)	Me = 20.0 IQR = 2.0	Me = 21.0 IQR = 2.0	Me = 19.0 IQR = 2.0	7.827 × 10^−11^
First dynamometer (*n_i_*)	JAMAR, *n* = 63 Camry, *n* = 71	JAMAR, *n* = 38 Camry, *n* = 32	JAMAR, *n* = 25 Camry, *n* = 39	0.299
Maximal strength _JAMAR_ (kg)	Me = 33.4 IQR = 12.6	Me = 40.0 IQR = 12.0	Me = 28.4 IQR = 7.8	4.387 × 10^−13^
Average strength _JAMAR_ (kg)	Me = 32.3 IQR = 12.1	Me = 32.0 IQR = 14.1	Me = 32.5 IQR = 10.2	1.205 × 10^−12^
ICC _JAMAR_	0.968	0.961	0.931	NC
SD differences _JAMAR_ (kg)	2.3	2.3	2.3	NC
SEM _JAMAR_ (kg)	0.4	0.4	0.6	NC
MDC _JAMAR_ (kg)	1.1	1.2	1.6	NC
Maximal strength _Camry_ (kg)	Me = 32.0 IQR = 13.1	Me = 39.0 IQR = 11.8	Me = 27.4 IQR = 7.6	9.127 × 10^−15^
Average strength _Camry_ (kg)	Me = 31.0 IQR = 12.6	Me = 37.3 IQR = 11.0	Me = 26.2 IQR = 6.98	8.215 × 10^−15^
ICC _Camry_	0.971	0.958	0.938	NC
SD differences _Camry_ (kg)	2.4	2.4	2.4	NC
SEM _Camry_ (kg)	0.4	0.5	0.6	NC
MDC _Camry_ (kg)	1.1	1.3	1.6	NC
Difference in maximal strength between devices (kg)	Me = 0.6 IQR = 2.9	Me = −0.1 IQR = 3.1	Me = 1.2 IQR = 2.5	0.377 × 10^−3^

Numerical values are expressed as absolute frequencies (*n*), percentages (%), medians (Me), interquartile ranges (IQRs), standard deviations (SDs), standard errors of measurements (SEMs), minimum detectable changes (MDCs), intraclass correlation coefficients (ICCs), and not computed (NC). *p*-values were only computed for the total sample in case of proportions (measuring side, first dynamometer), and the rest were computed grouping by sex.

**Table 3 nutrients-16-01824-t003:** Prevalence of dynapenia according to dynamometer type and cutoff point.

Cutoff Point	JAMAR Plus	Camry
Dodds et al. (p10) [45]	5 (3.67%)	7 (5.14%)
Dodds et al. (dichotomous) [45]	5 (3.67%)	6 (4.41%)

**Table 4 nutrients-16-01824-t004:** Basic anthropometric parameters and diagnosis of malnutrition in the study sample.

Parameter	Results
Weight (kg)	Me = 73.0 IQR = 18.2
Height (m)	Mean = 1.64 SD = 0.09
BMI (kg/m^2^)	Me = 27.3 IQR = 5.6
BMI by group (*n_i_*)	Underweight, *n* = 5 Normal weight, *n* = 44 Overweight, *n* = 50 Grade 1 obesity, *n* = 22 Grade 2 obesity, *n* = 8 Grade 3 obesity, *n* = 5
Percentage of weight loss (%)	Men: Me = −4.23; IQR = 12.3 Women: Me = −0.17; IQR = 3.37
GLIM malnutrition (*n_i_*)	No malnutrition, *n* = 94 Moderate malnutrition, *n* = 38 Severe malnutrition, *n* = 2

Basic anthropometric parameters and diagnosis of malnutrition. Numerical values are expressed as absolute frequencies (*n*), percentages (%), medians (Me), interquartile range (IQRs), mean, and standard deviation (SD). BMI: body mass index; GLIM: Global Leadership Initiative on Malnutrition.

**Table 5 nutrients-16-01824-t005:** BIA-derived parameters in the study sample.

Parameter	Men	Women	*p*-Value
R (Ω)	Mean = 503.0 SD = 67.9	Mean = 602.0 SD = 78.7	2.315 × 10^−12^
X_c_ (Ω)	Mean = 45.4 SD = 8.5	Mean = 50.6 SD = 9.4	0.105 × 10^−2^
Z (Ω)	Mean = 504.76 SD = 68.08	Mean = 604.4 SD = 79.0	5.336 × 10^−11^
θ (°)	Me = 5.3 IQR = 0.8	Me = 4.9 IQR = 0.7	0.874 × 10^−3^
Standardized θ (Z-score)	Me = −0.9 IQR = 1.3	Me = −0.8 IQR = 0.9	0.692
MM (kg)	Me = 26.6 IQR = 3.40	Me = 17.4 IQR = 3.93	<2.2 × 10^−16^
SMI (kg/m^2^)	Me = 9.60 IQR = 1.46	Me = 6.80 IQR = 1.19	<2.2 × 10^−16^
Muscle atrophy (*n_i_*)	Atrophy, *n* = 7 No atrophy, *n* = 62	Atrophy, *n* = 28 No atrophy, *n* = 36	0.890 × 10^−5^

Numerical values are expressed as absolute frequencies (n), medians (Mes), interquartile ranges (IQRs), means, and standard deviations (SDs). R: resistance; X_c_: reactance; θ: phase angle; MM: muscle mass; SMI: Skeletal Muscle Index. *p*-values were computed grouping by sex.

## Data Availability

The data presented in this study are available on request from the corresponding author due to European legislation on data protection.

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
