# Peer review of "A Cross-Sectional Validation Study of Camry EH101 versus JAMAR Plus Handheld Dynamometers in Colorectal Cancer Patients and Their Correlations with Bioelectrical Impedance and Nutritional Status"

_nutrients, 2024, doi:10.3390/nu16121824_

Round 1

Reviewer 1 Report

Comments and Suggestions for Authors

Thank you for submitting the manuscript "A Cross-Sectional Validation Study of Camry EH101 Versus JAMAR Plus Handheld Dynamometers in Colorectal Cancer Patients and Their Correlations With Bioelectrical Impedance and Nutritional Status" to Nutrients.

The manuscript describes the comparison between two force measurement devices. Although it is a valid comparison, the manuscript does not seem to fit the scope of Nutrients as it focuses on the devices and their efficiency and not on the population.

The manuscript is well written but the sample size still seems small and unrepresentative to say that there is efficiency in this device.

Author Response

Please see the attachment below

Reviewer 2 Report

Comments and Suggestions for Authors

This is an interesting reserch article with adequate novelty and quality. Some points should be addressed.

- In line 21, please delete "n=".

- In the consudion section, the authors should add more suggestions for what future studies could be performed based on the results of the present study.

- In line 37, there is a typos error as it is reported [13] as a reference. Please revise.

- The 2nd paragraph is a bit long. The authors should split this paragraph into 2 separate paragraphs.

- Subheading should be added in the Maaterials and Methods section.

- An institutional review board statement and an informed consent statement should also be included in the Materials and Methods section.

- The size of all Tables should be increased. Especially, the number and words of Tables should be a bit increased.

- Concerning the paragraph in lines 400-408, are there any published data concerning the topic of this paragraph in order to provide a comparison analysis with previous data?

- The last paragraph of the discusion could be presented as as separate Conclusion section.

Comments on the Quality of English Language

Minor editing of English language required

Reviewer 3 Report

Comments and Suggestions for Authors

The introduction is comprehensive, providing a detailed overview of the importance of nutritional screening, diagnosis, and intervention in oncology. It covers the relevance of muscle mass (MM) loss and various methods for measuring and estimating MM, particularly emphasizing bioelectrical impedance analysis (BIA) and its clinical applications. However, some aspects could be revised in order to increase readability. Particularly, the introduction is densely packed with information, which might be overwhelming for readers unfamiliar with the technical aspects of BIA and dynamometry. Simplifying or breaking down some of the more complex sections could enhance readability.

While detailed, the flow of information could be improved. The transition between different topics (e.g., from BIA to handgrip strength measurement) could be smoother to maintain a coherent narrative.

Lastrly, there is some redundancy in explaining the methods and their clinical applications. Streamlining these explanations could make the text more concise.

Lines 32-36: The emphasis on MM loss as an independent predictor of survival and other clinical outcomes is well-placed. However, consider briefly mentioning the prevalence of MM loss in cancer patients to contextualize its importance early on.

Lines 65-83: The discussion on handgrip strength measurement using dynamometers is detailed but could benefit from a brief introduction explaining why handgrip strength is a valuable measure in this context.

Methods are well described and detailed

Figures and graphs are self-explicative and well discussed.

Discussion: please consider to add a paragraph regarding implications of your results and another paragraph on future research

Round 2

Reviewer 1 Report

Comments and Suggestions for Authors

This reviewer is grateful for the explanations and corrections made to the text by the authors. They were of great importance in clarifying questions about the manuscript.